# Risk factors associated with student distress in medical school: Associations with faculty support and availability of wellbeing resources

**Simone Langness**[1][*], **Nikhil Rajapuram**[1], **Megan Marshall**[2], **Arifeen S. Rahman**[3], **Amanda Sammann**[1]

**1** Department of Surgery, University of California, San Francisco, San Francisco, California, United States of America, **2** New York University School of Medicine, New York, New York, United States of America, **3** Stanford University School of Medicine, Palo Alto, California, United States of America

☺ These authors contributed equally to this work.

* slangness@gmail.com

**Data Availability Statement:** Database uploaded in supplemental information. The names of medical schools and school city were deleted from the

## Abstract

### Background

It is estimated that over half of medical students experience severe distress, a condition that correlates with low mental quality-of-life, suicidal ideation and serious thoughts of dropping out. While several risk factors for the development of severe distress have been identified, most focus on individual student characteristics. Currently, little is known about the impact medical schools have on student wellbeing.

### Methods

Prospective, observational survey study from 2019–2020 from a national cohort of US medical students. Student wellbeing, school characteristics, and wellbeing resource availability was measured with a 30-question electronic survey. Medical student distress was defined as a Medical Student Wellbeing Index (MS-WBI) of $\geq 4$. Risk factors for the development of severe distress were evaluated in a multivariate logistic regression model. The impact of the number of wellbeing resources available on student wellbeing was measured along multiple wellbeing domains. Independent reviewers categorized free text analysis of survey responses about desired wellbeing resources into themes.

### Results

A total of 2,984 responses were included in the study, representing 45 unique medical schools. Medical school characteristics independently associated with severe distress included low faculty support (OR 4.24); the absence of mentorship resources (OR 1.63) and the absence of community building programs (OR 1.45) in a multivariate model. Increased availability of wellbeing resources was associated with lower average MS-WBI (4.58 vs. 3.19, p<0;05) and a smaller percentage of students who had taken or considered taking a

database to protect school identity. All other
school-specific data was kept in the database.

**Funding:** The author(s) received no specific
funding for this work.

**Competing interests:** The authors have declared
that no competing interests exist.

leave of absence (40% vs. 16%, p<0.05). The resources most desired by students were
mental health services and scheduling adjustments.

## Conclusions

The majority of medical school characteristic that contribute to student distress are modifiable. Improving faculty support and offering more and varied wellbeing resources may help to mitigate medical student distress. Student feedback is insightful and should be routinely incorporated by schools to guide wellbeing strategies.

## Introduction

Medical student wellbeing is at an all-time low. A recent study in a national cohorts of medical students demonstrated that over half of US medical students experience severe distress [1], a condition that correlates with low mental quality-of-life, recent suicidal ideation and/or serious thoughts of dropping out of school [2]. Depression amongst medical students has been reported between 9–55% [3], at least three times higher than age-matched college graduates [4]. Eleven percent of medical students have endorsed suicidal ideation [3], 44% suffer from burnout [5], and one in five has either taken or considered taking time off from medical school for their personal wellbeing [1].

Drivers of medical student wellbeing broadly fall into two categories: individual and institutional [6]. Individual risk factors are intrinsic student characteristics that predispose students to distress or exacerbate distress once encountered and include such elements as gender [7], debt burden [6, 8], and race [9, 10]. Non-male students (female and transgender), for example, are 60% more likely to experience distress than male students [1] and those with escalating debt burden have increasingly higher rates of distress, regardless of their chosen specialty [1, 8]. Other individual factors know to contribute to student distress include disability status [1], caretaker responsibilities [11] and the clinical phase in medical school [1, 6].

Institutional drivers of medical student wellbeing are direct products of medical schools and encompass such elements as curriculum, evaluations, and culture. Rohe et al, found that students from institutions who used a traditional letter grading system were twice as likely to suffer burnout as those taught under a pass/fail grading system [12, 13]. Other institutional risk factors for medical student distress include lack of control over one's schedule [14] and student mistreatment by residents and faculty [15–17]. Furthermore, there are several institutional factors that may adversely affect student wellbeing but have yet to be studied, such as university type (private versus public), geographic location and research emphasis, to name a few.

Additionally, schools can offer resources to positively affect student wellbeing. Studies evaluating institution-created wellbeing resources range considerably in effectiveness, are often of poor quality and may become ineffective after 6 months [18, 19]. As a result of this knowledge gap, schools vary significantly in the types of wellbeing resources they offer. Saint Louis University, for example, invested in a large curriculum overhaul that included such features as establishing learning communities, creating more flexible scheduling options, and increasing faculty contact hours to combat student burnout and depression [20]. Vanderbilt created a student wellness program that included a mentorship and advising program, a community building event, and a personal wellbeing class series [21]. Other institutions have invested in peer-mentorship programs [22], mindfulness training [23, 24], resilience programs [25], mental

health services [26, 27], and overall well-being education [21, 28] as strategies to combat medical student distress. The impact of these various strategies on a national level has yet to be evaluated. Furthermore, there is minimal data from students directly about which resources they value.

The goal of this mixed methods research study was to: a) holistically evaluate individual and institutional drivers of medical student distress, b) determine whether there are association between institutional investment in wellbeing resources and medical student wellbeing, and c) evaluate which wellbeing resources are most valued by students.

## Methods

### Study overview

This was a prospective observational study to evaluate the risk factors associated with medical student distress. Medical students were surveyed via an electronic questionnaire after institutional review board (IRB) approval was obtained from the University of California, San Francisco. Informed consent was obtained electronically prior to survey participation. The quantitative and qualitative data in this study is part of a larger research project on medical student wellbeing [1].

### Survey development & distribution

The Medical Student Wellbeing Survey (MSWS) was distributed to a national cohort of US medical students. Survey piloting and method of distribution have been previously described [1]. Briefly, a recruitment email was sent to a medical student liaison identified through the Association of American Medical Colleges at every US and Caribbean medical school. Liaison interested in the study distributing the MSWS to their own classmates. Additionally, access to the survey was available through the social media platforms Twitter and Facebook.

The MSWS included the Medical Student Wellbeing Index (MS-WBI), a comprehensive yet simple questionnaire to screen for student distress utilizing questions from the Maslach Burnout Index; the PRIME-MD depression test; and the Short Form-8 mental and physical quality of life screening [29]. An MS-WBI score ≥4 is associated with severe distress, correlating with low mental quality of life, recent suicidal ideation and/or serious thoughts of dropping out of school, with a sensitivity and specificity of 90% [2]. Additional components to the MSWS analyzed in this study included student's classification of their preclinical grading system and degree of faculty support as measured on a 3-part Likert scale (Not supportive at all, Somewhat supportive and Strongly supportive). Phase in medical school was self-identified by respondents and defined as "Pre-clinical" for time spent prior to, "Clinical" as time spent during, and "Post-Clinical" as time spent after completing core clinical clerkships. "Gap" was defined as dedicated time away from clinical work for research, additional degrees and/or to take a leave of absence. Only survey responses from allopathic medical schools were included in this study.

### Medical school characteristics

Respondents were asked to identify the medical school they were currently attending. Medical school names were then cross referenced against data published in the 2019 US News & World Report Medical School Ranking in order to extract school-specific data that may impact student wellbeing [30]. Data not available in the US News & World Report was gathered from published data on individual medical school websites. These data include: average tuition, preclinical duration, university type (private versus public), school location, research ranking, class size, faculty to student ratio, and average matriculant MCAT score. For public schools, the average tuition was

calculated as an average between the in-state and out-of-state tuition. Preclinical duration was defined as the time in medical school before starting core-clinical rotations with "standard" duration being ≥18 month and "abbreviated" being <18 months [31]. Class size terciles were calculated and defined as "large" for >1,000 students, "medium" for 500–1,000 students and "small" for <500 students. School location was further categorized by region and city size, as defined by the US Census Bureau using the following population definitions: large metropolitan (>1.5 million), metropolitan (500,000–1.5 million), medium-size urban area (200,000–500,000), small urban area (50,000–200,000) and urban clusters (2,500–50,000) [32].

## Wellbeing resources

Respondents were asked about the availability of common wellbeing resources cited in literature, including Mental Health & Counseling Services [33, 34], Peer Mentorship [35, 36], Self-Care Education [21, 37], Mindfulness/Meditation Classes [23, 24], and Community Building Events [38] at their medical school. Respondents were then asked about the availability of these resources and to specify which resources they had utilized. Lastly, respondents were given unlimited free text space to answer the prompt, "What wellbeing resource(s), if offered at your school, do you feel would be most beneficial?" The free text responses were independently reviewed by three researchers and inductively coded based on previously described techniques [39]. Unique themes and supporting quotes were identified and organized into pre-defined well-being domains [40]. Coding differences between reviewers were discussed until unanimous consensus was reached.

## Risk factor modeling

A multivariable logistic regression model was developed to determine risk factors for severe distress among survey respondents from allopathic medical schools. Individual student characteristics included in the model were medical school phase, age, gender, marital status, debt burden, underrepresented minority (URM) status, disability status, specialty competitiveness, and confidence in specialty. Specialty competitiveness categories were determined based on previously described methods [1]. The student's age was further categorized into ≥28 and <28 years-old to minimize cofounding effects of student's medical school phase.

Medical school characteristics included in the multivariable logistic regression model included university type, medical school classification, school region, school city population characteristic, preclinical grading system, class size, faculty to student ratio, faculty support, average matriculant MCAT score, average tuition, research ranking, and availability of wellbeing resources. Variables in the model were checked for multicollinearity using Variance Inflation Factor (VIF). All variables in the model had a VIF of <5 except for school region and top research quartile, which were both <6. A multivariate logistic regression analysis was also performed on medical school characteristics to determine predictors for having high resource availability. High resource availability was defined as three of more wellbeing resources. Responses from medical school wellbeing resource prediction analysis were only included if all data points were available (i.e., missing data was excluded). All statistical tests were two-sided and p <0.05 was considered significant. Statistical analyses were performed using SAS version 9.4 and R version 3.6.1.

## Results

### Survey respondents

A total of 2,984 responses were included in the study, representing 45 unique medical schools throughout the US. There was an even distribution of respondents from medical school phases

with 52.6% in the pre-clinical phase and 42.6% in the clinical/post-clinical phases. A small portion (4.9%) of respondents identified as being in a "gap" year. A majority of respondents identified as female (65.0%), between the ages of 22–27 (79.1%), and never married (86.3%). According to the definitions set forth by the AAMC, 10.5% of respondents were characterized as URM, including Black / African American, Hispanic / Latinx and Native American [41]. Nine percent of respondents identified as having a chronic disability or illness.

## Univariate risk factors for severe distress

On univariate analysis, several individual and institutional characteristics were associated with severe distress (Table 1). Individual risk factors included involvement in the clinical phase of medical school (62.0%) or gap year (62.8%) compared to preclinical year (49.9%, p<0.001), non-male gender (55.8 vs. 44.2%, p<0.001), disability status (64.4 vs. 35.6%, p<0.001), and confidence in specialty choice with moderate confidence being associated with severe distress (57.9 vs. 42.1%, p = 0.002). Institutional risk factors identified in this series included evaluations in a letter grade system (63.1 vs. 36.9%, p = 0.002), higher annual tuition (53.5 vs. 46.7%, p = 0.003), large class size (60.0 vs. 40.0%, p<0.001), unsupportive faculty (76.6 vs. 23.4%, p<0.001) and lower research ranking (59.8 vs. 40.2%, p<0.001). Medical schools located in the Midwest region were protective against severe distress (56.6 vs. 43.4%, p = 0.042). Respondents who did not identify the availability of the following wellbeing resources at their medical school were more likely to have severe distress compared to respondents who did: mental health resources (63.9 vs. 52.1%, p = 0.044), peer mentorship (65.8 vs. 49.4%, p<0.001), self-care education (59.6 vs. 46.9%, p<0.001), meditation/mindfulness classes (59.3 vs. 48.9%, p<0.001) and community building events (63.9 vs. 45.7%, p<0.001).

## Multivariate risk factors for severe distress

Results of the multivariable regression model are listed in Table 2. Controlling for all other variables in the model, severe distress was more likely for students in their clinical phase (OR 1.43, 95% CI 1.1–1.8, p = 0.01) and those in a gap year (OR 2.05, 95% CI 1.3–3.2, p<0.01) compared to students in their pre-clinical phase. Additional individual student variables significantly associated with severe distress included non-male gender (OR 1.38), disability (OR 1.78), higher debt burden (OR 2.13), and moderate confidence in selected specialty (OR 1.32), (95% CI and p-value listed in Table 2) The sole institutional risk factors associated with severe distress on multivariable regression model was faculty support. Compared to students who rated faculty as "strongly supportive," students who rated faculty as "not supportive at all" were more than four times as likely to have severe distress (OR 4.24, 95% CI 2.6–5.9, p<0.01) and twice as likely than those who rated faculty as "somewhat supportive" (OR 2.37, 95% CI 2.0–2.9, p<0.01). Lastly, severe distress was more common at schools where students did not have access to mentorship programs (OR 1.63, 95% CI 1.3–2.1, p<0.01) and community building events (OR 1.45, 95% CI 1.2–1.8, p<0.01).

## Wellbeing resource availability

A total of 2,886 respondents (97%) reported the availability of mental health and counseling services, and 1190 (41%) had utilized the resource (Fig 1). Community building events and peer mentorship were the most utilized wellbeing resources (51%) whereas meditation/mindfulness classes were the least utilized (27%).

Measurements of wellbeing correlated with the reported number of wellbeing resources available. The average MS-WBI for respondents who reported the availability of all five wellbeing resources was 3.19 ± 1.93 compared to 4.58 ± 1.85, 4.09 ± 1.86, and 3.78 ± 1.81 for one,

**Table 1. Impact of medical student and medical school characteristics on student distress (univariate model).**

| INDIVIDUAL CHARACTERISTICS | | | | MEDICAL SCHOOL CHARACTERISTICS | | | | WELLBEING RESOURCES | | | |
|---|---|---|---|---|---|---|---|---|---|---|---|
| | Severe Distress | No Distress | p-value | | Severe Distress | No Distress | p-value | | Severe Distress | No Distress | p-value |
| N (%) | 1570 (52.6%) | 1414 (47.4%) | | N (%) | 1554 (52.7%) | 1396 (47.3%) | | N (%) | 1503 (52.4%) | 1364 (47.6%) | |
| Medical School Year | | | | University Type | | | | Mental Health Resources | | | |
| Preclinical | 785 (49.9%) | 787 (50.1%) | **<0.001** | Private | 813 (52.8%) | 726 (47.2%) | 0.83 | Yes | 1450 (52.1%) | 1334 (47.9%) | **0.044** |
| Clinical | 380 (62.0%) | 233 (38.0%) | | Public | 757 (52.4%) | 688 (47.6%) | | No | 53 (63.9%) | 30 (36.1%) | |
| Post-Clinical | 314 (48.0%) | 340 (52.0%) | | School Region | | | | Peer Mentorship | | | |
| Gap | 91 (62.8%) | 54 (37.2%) | | Midwest | 431 (43.4%) | 331 (56.6%) | **0.042** | Yes | 1151 (49.4%) | 1181 (50.6%) | **<0.001** |
| Age | | | | Northeast | 570 (49.4%) | 556 (50.6%) | | No | 352 (65.8%) | 183 (34.2%) | |
| <28 | 1258 (52.3%) | 1146 (47.7%) | 0.48 | South | 163 (49.7%) | 127 (50.3%) | | Self Care Education | | | |
| ≥28 | 300 (54.1%) | 255 (45.9%) | | West | 395 (50.0%) | 391 (50.0%) | | Yes | 763 (46.9%) | 863 (53.1%) | **<0.001** |
| Gender | | | | City characteristic | | | | No | 740 (59.6%) | 501 (40.4%) | |
| Male | 468 (46.6%) | 536 (53.4%) | **<0.001** | Large Metropolitan | 538 (53.8%) | 462 (46.2%) | 0.74 | Meditation/Mindfulness | | | |
| Non-Male | 1083 (55.8%) | 859 (44.2%) | | Medium-size urban areas | 131 (53.3%) | 115 (46.7%) | | Yes | 926 (48.9%) | 968 (51.1%) | **<0.001** |
| URM | 186 (59.0%) | 129 (41.0%) | | Metropolitan | 324 (50.8%) | 314 (49.2%) | | No | 577 (59.3%) | 396 (40.7%) | |
| Disability | 170 (64.4%) | 94 (35.6%) | **<0.001** | Small urban area | 400 (51.7%) | 373 (48.3%) | | Community Build | | | |
| Marital Status | | | | Urban Clusters | 177 (54.1%) | 150 (45.9%) | | Yes | 824 (45.7%) | 980 (54.3%) | **<0.001** |
| Never Married | 1353 (52.5%) | 1222 (47.5%) | 0.96 | Grading System | | | | No | 679 (63.9%) | 384 (36.1%) | |
| Married | 194 (53.4%) | 169 (46.6%) | | Letter Grades (A, B, C, etc.) | 99 (63.1%) | 58 (36.9%) | **0.002** | | | | |
| Divorced/Widowed | 10 (52.6%) | 9 (47.4%) | | Other: | 6 (75.0%) | 2 (25.0%) | | | | | |
| Debt | | | | Pass/Fail + Honors/High Pass | 318 (55.9%) | 251 (44.1%) | | | | | |
| <$20K | 388 (43.4%) | 506 (56.6%) | **<0.001** | Pass/Fail | 1070 (50.7%) | 1041 (49.3%) | | | | | |
| $20K-$100K | 469 (51.8%) | 436 (48.2%) | | Pre-clinical duration | | | | | | | |
| $100-$300K | 524 (60.6%) | 340 (39.4%) | | Abbreviated | 645 (50.6%) | 629 (49.4%) | 0.086 | | | | |
| >$300K | 85 (67.5%) | 41 (32.5%) | | Standard | 802 (53.9%) | 685 (46.1%) | | | | | |
| I'm Not Sure | 49 (58.3%) | 35 (41.7%) | | Class size | | | | | | | |
| Specialty Competitiveness | | | | Large | 317 (60.0%) | 211 (40.0%) | **<0.001** | | | | |
| Low | 720 (51.1%) | 690 (47.1%) | 0.22 | Medium | 967 (52.1%) | 890 (47.9%) | | | | | |
| Moderate | 559 (54.6%) | 464 (45.4%) | | Small | 270 (47.8%) | 295 (52.2%) | | | | | |
| High | 249 (52.9%) | 222 (47.1%) | | Faculty:Student Ratio | 2.11 ± 1.66 | 2.48 ± 1.96 | **<0.001** | | | | |
| Confidence in Specialty | | | | Faculty Support | | | | | | | |
| Low | 284 (48.3%) | 304 (51.7%) | **0.002** | Not supportive at all | 151 (76.6%) | 46 (23.4%) | **<0.001** | | | | |
| Moderate | 389 (57.9%) | 283 (42.1%) | | Somewhat supportive | 933 (60.8%) | 602 (39.2%) | | | | | |
| High | 885 (52.2%) | 809 (47.8%) | | Strongly supportive | 409 (36.7%) | 704 (63.3%) | | | | | |
| | | | | Average yearly tuition | | | | | | | |
| | | | | <$40K | 174 (44.6%) | 216 (55.4%) | **0.003** | | | | |
| | | | | $40-60K | 1011 (54.1%) | 857 (45.9%) | | | | | |
| | | | | >$60K | 369 (53.3%) | 323 (46.7%) | | | | | |
| | | | | Research ranking | | | | | | | |
| | | | | Q1 | 593 (47.9%) | 645 (52.1%) | **<0.001** | | | | |
| | | | | Q2 | 265 (52.1%) | 244 (47.9%) | | | | | |
| | | | | Q3 | 269 (54.2%) | 227 (45.8%) | | | | | |
| | | | | Q4 | 443 (59.8%) | 298 (40.2%) | | | | | |
| | | | | Avg Matriculant MCAT Score | 146.7 ± 14.2 | 135.7 ± 20.3 | 0.69 | | | | |

URM = student from underrepresented minority, Q1 = top quartile, Q2 = 2nd quartile, Q3 = 3rd quartile, Q4 = bottom quartile.

two and three resources, respectively (p<0.01 for all comparisons). The percentage of respondents who reported having taken or considered taking a leave of absence for personal wellbeing was 40% for those who reported one available wellbeing resource compared to 16% for those who reported the availability of all five wellbeing resources (p<0.01). Respondents who reported more wellbeing resources also had less negative change to their physical, emotional, and social wellbeing since starting medical school compared to those who reported fewer wellbeing resources. There was no difference between wellbeing measurements for respondents reporting four available wellbeing resources compared to those reporting five except for

**Table 2. Impact of medical school characteristics on student distress (multivariate model).**

| | OR (95% CI) | p-value | | OR (95% CI) | p-value | | OR (95% CI) | p-value |
|---|---|---|---|---|---|---|---|---|
| **INDIVIDUAL CHARACTERISTICS** | | | **MEDICAL SCHOOL CHARACTERISTICS** | | | **WELLBEING RESOURCES** | | |
| **Medical School Year (vs. Preclinical)** | | | **University Type (vs. Public)** | | | **(vs. Available)** | | |
| Clinical | 1.43 (1.11–1.84) | **0.01** | Private | 1.07 (0.75–1.53) | 0.71 | **Mental Health Resources** | 1.01 (0.48–2.12) | 0.98 |
| Post-Clinical | 0.76 (0.57–1) | **0.05** | **Medical School Region (vs. South)** | | | **Mentorship** | 1.63 (1.25–2.13) | **<0.01** |
| Gap | 2.05 (1.31–3.22) | **<0.01** | Midwest | 1.24 (0.75–2.03) | 0.4 | **Self Care Education** | 1.15 (0.94–1.4) | 0.19 |
| **Age (vs. <28)** | | | Northeast | 1.26 (0.82–1.93) | 0.29 | **Meditation/Mindfulness** | 0.95 (0.77–1.17) | 0.64 |
| ≥28 | 0.85 (0.67–1.09) | 0.21 | West | 1.09 (0.7–1.72) | 0.7 | **Community Building Events** | 1.45 (1.17–1.79) | **<0.01** |
| **Gender (vs. Male)** | | | **City characteristic (vs. Large Metropolitan** | | | | | |
| Non-Male | 1.38 (1.14–1.67) | **<0.01** | Medium-size urban areas | 1.08 (0.67–1.74) | 0.76 | | | |
| **Marital Status (vs. Never Married)** | | | Metropolitan | 1.02 (0.69–1.52) | 0.92 | | | |
| Divorced/Widowed | 0.79 (0.24–2.6) | 0.7 | Small urban area | 1.09 (0.77–1.55) | 0.62 | | | |
| Married | 1.1 (0.84–1.46) | 0.49 | Urban Clusters | 0.69 (0.39–1.02) | 0.09 | | | |
| **Debt (vs. <$20K)** | | | **Grading System (vs. Pass/Fail)** | | | | | |
| $20K-$100K | 1.42 (1.14–1.78) | **<0.01** | Letter Grades (A, B, C, etc.) | 0.8 (0.36–1.77) | 0.58 | | | |
| $100-$300K | 1.61 (1.27–2.04) | **<0.01** | Other: | 2.12 (0.36–12.64) | 0.41 | | | |
| >$300K | 2.13 (1.25–3.66) | **<0.01** | Pass/Fail + Honors/High Pass | 0.97 (0.76–1.25) | 0.82 | | | |
| I'm Not Sure | 1.56 (0.9–2.7) | 0.11 | **Pre-clinical duration (vs. Standard)** | | | | | |
| **URM** | 1.28 (0.96–1.73) | 0.1 | Abbreviated | 1.33 (1–1.77) | 0.06 | | | |
| **Disability** | 1.78 (1.31–2.43) | **<0.01** | **Class size (vs. Small)** | | | | | |
| **Specialty Competitiveness (vs. Low)** | | | Large | 1.06 (0.63–1.78) | 0.83 | | | |
| High | 1.24 (0.96–1.59) | 0.1 | Medium | 1.06 (0.8–1.39) | 0.7 | | | |
| Moderate | 1.13 (0.93–1.37) | 0.23 | **Faculty:Student Ratio** | 0.93 (0.85–1.01) | 0.1 | | | |
| **Confidence in Specialty (vs. High)** | | | **Faculty Support (vs. Strongly Supportive)** | | | | | |
| Low | 0.94 (0.74–1.19) | 0.6 | Not supportive at all | 4.24 (2.61–6.88) | **<0.01** | | | |
| Moderate | 1.32 (1.05–1.67) | **0.02** | Somewhat supportive | 2.37 (1.96–2.87) | **<0.01** | | | |
| | | | **Average Matriculant MCAT Score** | 1.05 (0.7–1.57) | 0.82 | | | |
| | | | **Average tuition (vs. <$40K)** | | | | | |
| | | | $40-60K | 1.3 (0.87–1.96) | 0.2 | | | |
| | | | >$60K | 1.04 (0.7–1.55) | 0.83 | | | |
| | | | **Research Ranking (vs. Q4)** | | | | | |
| | | | Q1 | 0.68 (0.42–1.09) | 0.11 | | | |
| | | | Q2 | 0.85 (0.54–1.34) | 0.48 | | | |
| | | | Q3 | 0.84 (0.56–1.27) | 0.41 | | | |

URM = student from underrepresented minority, Q1 = top quartile, Q2 = 2nd quartile, Q3 = 3rd quartile.

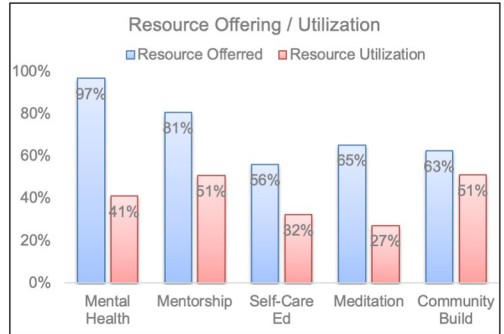
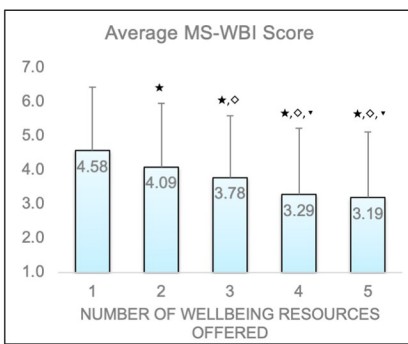
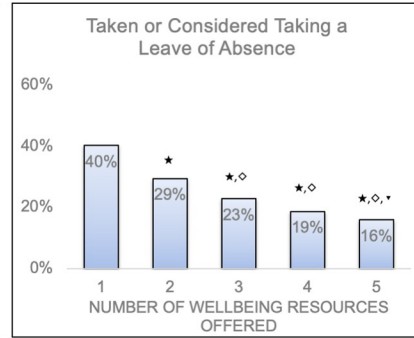

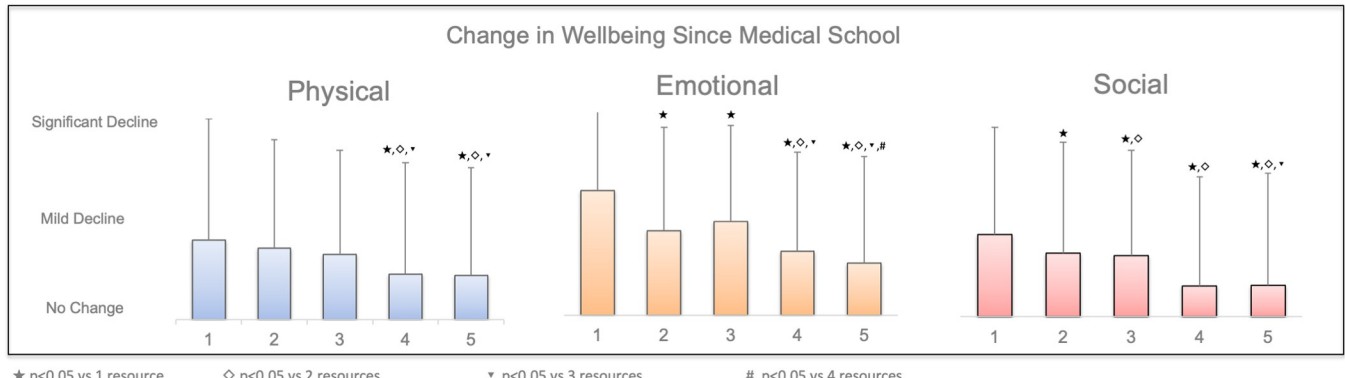

★ p<0.05 vs 1 resource     ◇ p<0.05 vs 2 resources     ▾ p<0.05 vs 3 resources     # p<0.05 vs 4 resources

**Fig 1. Impact of number of wellbeing resources offered on student distress.** Availability and utilization of well-being resources (Mental Health & Counseling Services, Peer Mentorship, Self-Care Education, Mindfulness/Meditation Classes, and Community Building Events Mental Health) at US allopathic medical schools as reported by medical student survey respondents. Average MS-WBI and percentage of students who had taken or considered taking a leave of absence for personal wellbeing based on the number of reported wellbeing resources offered. Average change in students' physical, emotional, and mental well-being based on the number of wellbeing resources offered. Error bars represent standard deviation, p-value vs. 1, 2, 3 and 4 wellbeing resources noted in key.

change in emotional wellbeing since medical schools (-0.63 ± 0.98 versus -0.51 ± 1.05, p = 0.02). There was no consistent correlation between the percentage of resources utilized and average MS-WBI, percentage of students who had taken or considered taking a leave of absence or change in physical, emotional, or social wellbeing since starting medical school (Fig 2).

## Medical school predictors of high wellbeing resources

A total of 2,670 responses were included in the medical school wellbeing resource predictor analysis. Controlling for all other variables, public universities were less likely to have high wellbeing resource than private universities (OR 0.37, 95% CI 0.23–0.58, p<0.01) as were medical schools with medium class sizes (OR 0.59, 95% CI 0.36–0.95, p = 0.03) compared to large class size (Table 3). Medical schools with lower average tuition (<$40K or $40-60K) were more likely to have high wellbeing resources available (OR 1.81, 95% CI 1.01–3.31, p<0.01 and OR 1.63, 95% CI 1.06–2.53), p < 0.01, respectively) compared to schools with higher average tuition (>$60K). The strongest predictor of high wellbeing resource availability was having strongly supportive faculty (OR 11.8, 95% CI 7.4–19.19, p<0.01) and somewhat supportive faculty (OR 3.28, 95% CI 2.17–4.95, p<0.01) compared to faculty who are not supportive at all.

## Desired wellbeing resources

A total of 1,054 respondents provided free text answers to the prompt: "What wellbeing resource(s), if offered at your school, do you feel would be most beneficial?" Most of the

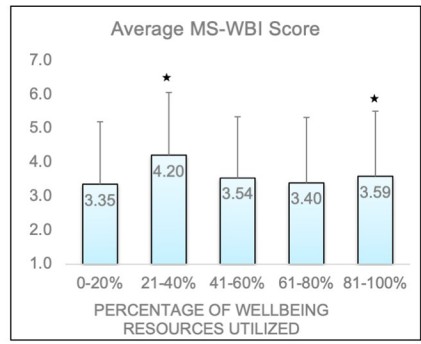
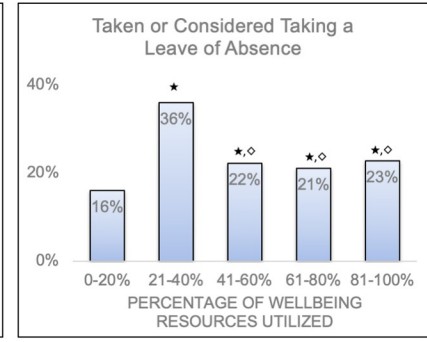

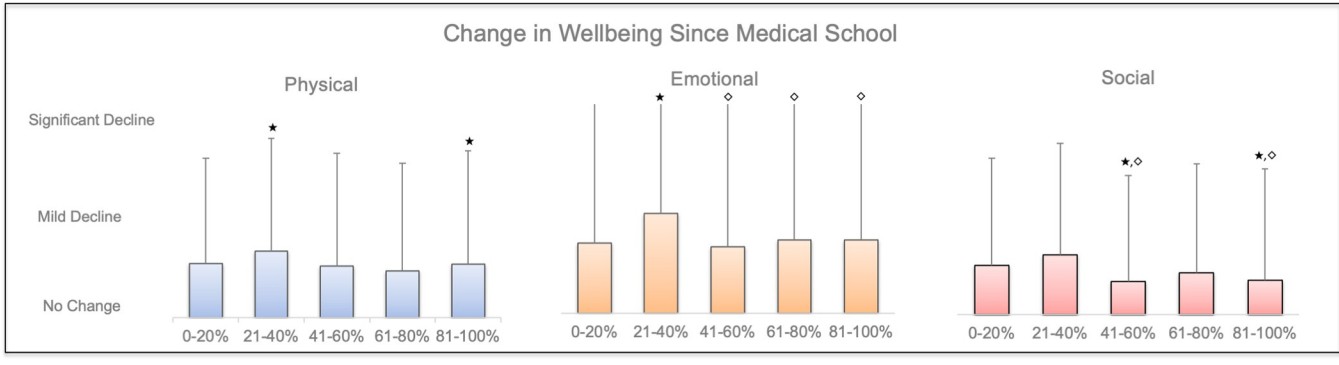

**Fig 2. Impact of wellbeing resources utilization on student distress.** Average MS-WBI and percentage of students who had taken or considered taking a leave of absence for personal wellbeing based on self-reported utilization rate of wellbeing resources (Mental Health & Counseling Services, Peer Mentorship, Self-Care Education, Mindfulness/Meditation Classes, and Community Building Events Mental Health) by survey respondents. Utilization rates were calculated based only on resources marked as offered by students. Average change in students' physical, emotional, and mental well-being based on the number of wellbeing resources offered. Error bars represent standard deviation, p-value vs. 1, 2, 3 and 4 wellbeing resources noted in key.

suggestions (447, 32%) were categorized into the emotional domain of wellness, followed by school/career (358, 26), social (241, 17%), physical (171, 12%), and financial (120, 9%) (Fig 3A). The most desired wellbeing resource was Mental Health Services (346, 25%). Frequently encountered themes for Mental Health Services included limited access to current mental health services, inadequate time to utilize current services, desire for dedicated counselors specialized in working with medical students, longer duration of counseling offered and policy changes that would make mental health counseling an opt-out default for students rather than opt-in. Resource themes and representative quotes are listed in Fig 3B.

Wellbeing resources aimed at adjusting the medical student schedule were the second most requested (194, 14%). Most answers centered around a desire for more scheduled time off with variations on having specific time off during weekday working hours, time off between rotations and/or exams or being free from assignments or expectations during given time off. Other common themes included the desire for better mechanisms to request time off where the stigma and fear of the request was minimized and the desire for better control, flexibility, and advanced warning of schedules in order to plan for important events or schedule time to utilize wellbeing resources.

## Discussion

Our study is the first of its kind to holistically evaluate individual and institutional factors affecting on medical student wellbeing. In doing so, we found that medical schools may both contribute to and help mitigate against severe distress. We confirmed previous studies

**Table 3. Medical school predictors of high ($\geq$ 3) wellbeing resource availability.**

| | OR (95% CI) | p-value |
|---|---|---|
| **MEDICAL SCHOOL CHARACTERISTICS** | | |
| **University Type (vs. Private)** | | |
| Public | 0.37 (0.23–0.58) | **<0.01** |
| **Medical School Region (vs. Midwest)** | | |
| South | 0.61 (0.33–1.15) | 0.13 |
| Northeast | 0.55 (0.34–1.05) | 0.08 |
| West | 0.95 (0.56–1.59) | 0.84 |
| **City characteristic (vs. Large Metropolitan** | | |
| Medium-size urban areas | 1.01 (0.55–1.86) | 0.99 |
| Metropolitan | 0.74 (0.43–1.22) | 0.24 |
| Small urban area | 1.28 (0.76–2.16) | 0.35 |
| Urban Clusters | 0.72 (0.39–1.32) | 0.28 |
| **Grading System (vs. Letter Grades)** | | |
| Pass/Fail | 0.49 (0.13–1.47) | 0.24 |
| Pass/Fail + Honors/High Pass | 0.58 (0.16–1.76) | 0.38 |
| **Pre-clinical duration (vs. Standard)** | | |
| Abbreviated | 0.87 (0.59–1.29) | 0.49 |
| **Class size (vs. Large)** | | |
| Small | 0.74 (0.39–1.38) | 0.35 |
| Medium | 0.59 (0.36–0.95) | **0.03** |
| **Faculty:Student Ratio** | 1.12 (0.99–1.30) | 0.11 |
| **Faculty Support (vs. Not supportive at all)** | | |
| Somewhat supportive | 3.28 (2.17–4.95) | **<0.01** |
| Strongly supportive | 11.8 (7.40–19.19) | **<0.01** |
| **Average tuition (vs. >$60K)** | | |
| <$40K | 1.81 (1.01–3.31) | **0.05** |
| $40–60K | 1.63 (1.06–2.53) | **0.03** |
| **Research Ranking (vs. Q1)** | | |
| Q2 | 1.32 (0.82–2.14) | 0.25 |
| Q3 | 1.14 (0.66–2.01) | 0.64 |
| Q4 | 1.25 (0.66–2.41) | 0.50 |

Q1 = top quartile, Q2 = 2nd quartile, Q3 = 3rd quartile.

demonstrating that gender, phase in medical school, debt burden and disability status contribute to student distress while also discovering that the largest driver of severe distress is poor faculty support. Additionally, we found that school can be protective against severe distress by offering a variety of wellbeing resources. These findings reveal critical insights that can be used to help guide institutions on how best to support medical student wellbeing.

For schools interested in curbing medical student distress, the results from this study are encouraging as they reveal that the medical school characteristics that contribute to student distress are modifiable. Through actions like restructuring curriculum or student schedules, retraining faculty, and repurposing funding to invest in wellbeing resources, medical schools can create a healthier environment for students to thrive. Furthermore, as the scale of these actions can range from simple to complex, all medical schools can participate in improving student wellbeing.

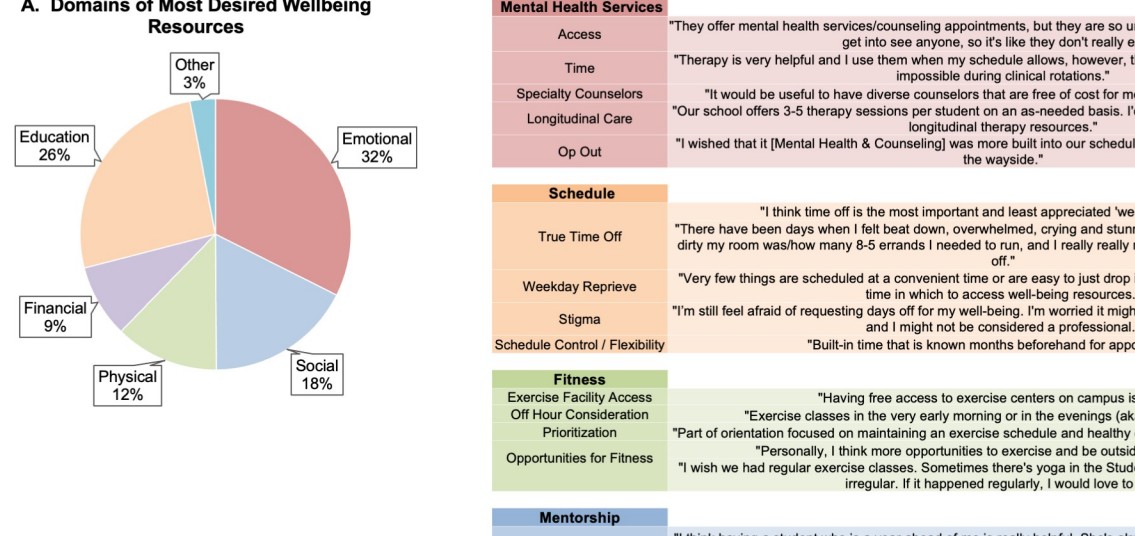

**Fig 3. Most desired wellbeing resources by students.** Characterization of student free-text responses according to domains of wellbeing (physical, emotional, social, educational, financial, and other) (A). Domain themes and representative quotes based of desired wellbeing resources (B).

## Wellbeing resources

In our study, having more wellbeing resources at one's school was associated with less severe distress. Interestingly, the availability of multiple wellbeing resources moderated the relationship with severe distress regardless of whether students utilized the resource. Several variables may help to explain this discrepancy. The first possibility is that by having more wellbeing resources available to students, medical schools are establishing an organizational culture that prioritizes wellbeing. Additionally, by having a diverse array of wellbeing resources, medical schools may help to normalize the reality of stress for students and lessen the stigma of depression and burnout, which only further drives distress [6, 42]. By avoiding stigma, students may avoid the compounding effects of depression and burnout to the routine stress endured within medical school, preventing the need to access these services as frequently.

We hope these data help institutions in the evaluation of their own programs and guide investment in new programs and resources. For example, mentorship programs, either faculty-student or student-student mentoring, can be implemented at most institutions with only a modest amount of organization, guidance, and motivation [35, 36]. Community building activities, such as group meals, events or coordinated volunteering efforts, can also be readily adapted into medical school culture without large budget or wide-spread curricular changes.

## Faculty development

Non-supportive faculty was the most predictive factor for the presence of severe distress in medical students (OR 4.2) in our study. The correlation between poor faculty support and

student depression, anxiety and attrition are paralleled in literature from other graduate student programs [43, 44]. In many ways, faculty represent the backbone of an institution as the conduits of knowledge for students. Faculty members are in a unique position to not only teach students but also to inspire, guide, and mentor them in their journey to becoming a future colleague. Collectively, faculty have the greatest impact on an institution's culture and values. Perhaps it is not surprising then that faculty have the greatest influence on student wellbeing.

Faculty support and engagement can be improved through the use of faculty development programs [45]. Pandachuck et al. found a significant improvement in student overall opinion of faculty members as instructors and in students feeling respected by faculty who participated in a teaching enhancement workshop [46]. Branch et al. found improvement in faculty humanistic teaching after completing a longitudinal development course [47]. Given the significant impact that faculty support has on medical student wellbeing, medical schools should consider the adoption of programs and policies that support faculty mentorship training and mitigate additional faculty burdens that deter from student support.

## Defining faculty support

One challenge to improving faculty support is that there is no clear definition of what supportive means to medical students. For some students, faculty support may mean frequent engagement, while for others, it may mean mentorship, interest in students' personal wellbeing or a willingness to extend themselves to help ensure student success. A more thorough understanding of student perception of faculty support deserves further attention. We are in the process of investigating this topic through a mixed-method approach as well as studying the factors that prevent faculty from being more supportive.

At a minimum, supportive faculty should create a learning environment that is free of discrimination and harassment, which may be less ubiquitous than previously thought. In a recent study of doctoral students from various fields, 16% reported having experienced sexual harassment, a number that increased to 21.5% in a female-only cohort [48]. In a national cohort of senior medical students, 86.7% reported mistreatment in the form of public humiliation, 26.4% reported being threatened with physical harm and 55% felt they had been sexually harassed [49]. Furthermore, medical student mistreatment has been associated with higher rates of depression, anxiety, burnout, dropping out and suicide [17, 50].

## Listening to students

Ultimately, institutional guidance on how best to prevent medical student distress needs to involve students themselves. Our study is the first to qualitatively assess what wellbeing resources students' value and believe will be most helpful to them. When provided a chance to give free text commentary, over 1,000 students (>1/3 of respondents) provided rich insights into both novel resources that would impact their wellbeing and structural barriers to prevent use of current resources and/or minimize their feelings of helpfulness.

Through our qualitative data, we learned that students' value and desire expansion of their currently available mental health services. We also learned that simply offering mental health services to medical students is inadequate if they cannot easily schedule appointments in a reasonable time frame, access services due to inflexible schedules or do not have counselors that can understand their unique stressors. These qualitative findings are critical for creating actionable and meaningful change to student wellbeing that would otherwise be lost in traditional quantitative surveys.

Our study also found that over a quarter of students requested an adjustment in their schedule to allow for more time off and for greater control and flexibility of their schedules as their most desired wellbeing resources. Excessive workload demands play heavily into medical student wellbeing. In a recent study, academic workload was the most frequently cited stressors in a national cohort of medical students [14].

No data currently exists on the average work hours for preclinical or clinical medical students. Most institutions, however, have adopted policies restricting medical student duty hours during clinical rotations to be like those set for resident physicians (no more than 80 hours / week, no more than 24 hours of continuous on-site duty). Clinical medical students do bear additional workload demands that often take place outside of scheduled duty hours, such as studying for their shelf exams, participation in research and organizing their applications for residencies. Furthermore, students are rarely given advance warning of their schedule, thus limiting their ability to plan for routine activities such as going to doctor or utilizing wellbeing resources or to participate in important family and community events. Lastly, many clinical rotations do not effectively utilize medical student time to optimize for education, especially considering research which suggests that additional time on clinical rotations does not necessarily translate to a better education [51, 52]. When students are pushed to stay long hours that are not felt to be educational, while being acutely aware of the elements of their lives that aren't being addressed, resentment and depression can thrive. As one student in our study stated, "Nothing, literally nothing is helpful except time off. We need time to go to the doctor, we need time to go to the dentist, we need time to exercise. We should take even a minute clue from tech and realize that well-paid, well-rested, well-treated individuals are more productive, more engaged, and do better work." Medical student time should be spent meaningfully and with consideration for the negative effects that prolonged and unstructured time on rotations can have on mental wellbeing. Efforts to standardize the release of student schedules and to provide options for greater flexibility and off times are likely to provide meaningful advances in student wellbeing.

## Study limitations

There are several limitations to our study. Due to our survey distribution method, we were unable to calculate a survey response rate. Additionally, there is a potential for response bias as there was a higher percentage of female students who responded to the survey than male (65.9% vs 34.1%). Both of these concerns have been addressed in our previous paper on medical student wellbeing [1]. A third limitation to this study is that we evaluated the effect of wellbeing resources of student distress based on student's perception of the resource availability and not on an objective determination of whether a school has a particular wellbeing resource. It is important to note that there is not uniformity amongst answers for available wellbeing resources from students at the same institutions. On average, student congruency on resource availability at a given school was as follows: Mental Health 94% (71–100%), Mentorship 76% (44–100%), Self-Care Education 48% (25–100%), Meditation 55% (20–100%) and Community Build 59% (32–100%). Using student perception of available resources was chosen as there is currently no database that exists describing available wellbeing resources for all medical schools in the US. Furthermore, we believe that knowing whether a student is aware of a given resource is valuable information for understanding student wellbeing. We are currently in the process of evaluating a more objective method for determine wellbeing resource availability at medical schools as well as understanding the factors at large that drive students to know and use their local wellbeing resource availability.

## Conclusion

Our mixed-methods study from a large cohort of U.S. medical students demonstrates that medical schools may themselves play a role in medical student wellbeing, specifically through faculty support and the available of wellbeing resources. Schools must take ownership in their role for student distress instead of focusing on individual student risk factors. Schools would benefit from investment more into creating a supportive faculty community, offering more and varied wellbeing resources, providing students with more control and flexibility in their schedules and with incorporating student feedback and suggestions into their wellbeing action plans.

## Supporting information

**S1 File.**
(XLSX)

## Author Contributions

**Conceptualization:** Simone Langness, Nikhil Rajapuram.

**Data curation:** Simone Langness, Nikhil Rajapuram, Megan Marshall, Arifeen S. Rahman.

**Formal analysis:** Simone Langness, Nikhil Rajapuram, Megan Marshall, Arifeen S. Rahman.

**Methodology:** Simone Langness.

**Supervision:** Amanda Sammann.

**Writing – original draft:** Simone Langness.

**Writing – review & editing:** Simone Langness, Nikhil Rajapuram, Amanda Sammann.

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
