## [Decision Letter · Decision Letter 0]

9 Jun 2021

PONE-D-20-40527

Medical Schools Contribute to Student Distress:The Impact of Faculty Support and Wellbeing Resource Availability

PLOS ONE

Dear Dr. Langness,

Thank you for submitting your manuscript to PLOS ONE. After careful consideration, we feel that it has merit but does not fully meet PLOS ONE’s publication criteria as it currently stands. Therefore, we invite you to submit a revised version of the manuscript that addresses the points raised during the review process.

We look forward to receiving your revised manuscript.

Kind regards,

Kamran Sattar

Academic Editor

PLOS ONE

Journal Requirements:

When submitting your revision, we need you to address these additional requirements

2. Please note that in order to use the direct billing option the corresponding author must be affiliated with the chosen institute. Please either amend your manuscript to change the affiliation or corresponding author, or email us at plosone@plos.org with a request to remove this option.

3.PLOS requires an ORCID iD for the corresponding author in Editorial Manager on papers submitted after December 6th, 2016. Please ensure that you have an ORCID iD and that it is validated in Editorial Manager. To do this, go to ‘Update my Information’ (in the upper left-hand corner of the main menu), and click on the Fetch/Validate link next to the ORCID field. This will take you to the ORCID site and allow you to create a new iD or authenticate a pre-existing iD in Editorial Manager. Please see the following video for instructions on linking an ORCID iD to your Editorial Manager account: https://www.youtube.com/watch?v=_xcclfuvtxQ.

Additional Editor Comments (if provided):

Kindly read reviewers' comments very carefully and do the necessary. Thanks

Reviewers' comments:

Reviewer's Responses to Questions

**Comments to the Author**

1. Is the manuscript technically sound, and do the data support the conclusions?

Reviewer #1: Yes

Reviewer #2: Partly

2. Has the statistical analysis been performed appropriately and rigorously? 

Reviewer #1: Yes

Reviewer #2: Yes

3. Have the authors made all data underlying the findings in their manuscript fully available?

Reviewer #1: Yes

Reviewer #2: Yes

4. Is the manuscript presented in an intelligible fashion and written in standard English?

Reviewer #1: Yes

Reviewer #2: Yes

5. Review Comments to the Author

Reviewer #1: The study is very interesting and very rich.

There are modifications to be made to the bibliography and details to be added

Introduction

References 3, 4 and 5 should be removed because they are old and provide less information than reference 7 (Rotenstein et al, 2016) which is more recent and better because it is a very good meta-analysis which takes into account the problem of the heterogeneity of scales and cutoffs.

There is also the Mata et al 2015 meta-analysis for interns (Prevalence of Depression and Depressive Symptoms Among Resident Physicians). If the authors wish to raise the issue of burnout, a meta-analysis has also been published (Frajerman et al, 2019 Burnout in Medical students before residency)

"To date, research on medical student wellbeing has overwhelmingly focused on 95 individual, not institutional factors ”This sentence is exaggerated when we know there is a meta-analysis on the subject (Wasson et al, 2016 Association Between Learning Environment Interventions and Medical Student Well-being A Systematic Review).

On interventions to improve student well-being, it should be noted that most of the studies are of low quality.

In addition, the effectiveness of individual interventions tends to disappear after 6 months unlike institutional interventions (Frajerman, 2020 Which interventions improve the well-being of medical students? A review of the Literature)

Methods

Even if protocol was described in a previous article, authors have to put the bases in this article: the period of the study and how the questionnaire was sent to justify absence of response rate

Before performing the multivariate analyzes, it would be interesting to look at certain correlations, in particular between MEDICAL SCHOOL CHARACTERISTICS and WELLBEING RESOURCES.

It is not clear to me whether "non male gender" represents women or women + transgender and non-binary. If it's just women, you better write it clearly.

The cut off for age seems strange and very high to me: why not have done 3 groups as for the course. This would be all the more interesting for multivariate analyzes

Marital status is also strange: being in a couple is protective, but you can be in a couple without being married.

The part “change in wellbeing since medical school” in figures 2 and 3 seems very questionable to me because it mixes up students of all levels. A student who feels very bad in 2nd year and only bad in 3rd year to consider that he There has been an improvement, but if, on the other hand, they are all asked to recall closely to before entering medicine, there is an obvious recall bias related to their seniority.

It would be more judicious to stratify by level of course or else to leave nothing.

Discussion

The risk factors identified are not specific to medical universities.

Authors wrote "especially in light of recent research which suggests that additional time on clinical rotations does not necessarily translate to a better education [51, 52]."

You cant's use the word recent for studies published in 2006 and 2001

Reviewer #2: Manuscript # PONE-D-20-40527 Title: Medical Schools Contribute to Student Distress: The Impact of Faculty Support and Wellbeing Resource Availability

Line 107 (Before methodology ) The goal of this mixed methods research study was to: a) holistically evaluate individual 107

and institutional drivers of medical student distress, b) determine the............

Line 431 (Conclusion) Our mixed-methods study from a large cohort of U.S. medical students demonstrates that

medical schools themselves play a role in medical student wellbeing

But there is lack of Qualitative data , authors should clarify this point.

6. PLOS authors have the option to publish the peer review history of their article (what does this mean?). If published, this will include your full peer review and any attached files.

Reviewer #1: **Yes: **Ariel FRAJERMAN

Reviewer #2: No

---

## [Author Response · Author response to Decision Letter 0]

13 Oct 2021

Introduction

1. Reference Update.

“References 3, 4 and 5 should be removed because they are old and provide less information than reference 7 (Rotenstein et al, 2016) which is more recent and better because it is a very good meta-analysis which takes into account the problem of the heterogeneity of scales and cutoffs.”

“There is also the Mata et al 2015 meta-analysis for interns (Prevalence of Depression and Depressive Symptoms Among Resident Physicians). If the authors wish to raise the issue of burnout, a meta-analysis has also been published (Frajerman et al, 2019 Burnout in Medical students before residency)”

We appreciate the reviewer bringing these additional studies to our attention. The references have been adjusted as suggested.

2. Individual versus Institutional Factors

"To date, research on medical student wellbeing has overwhelmingly focused on individual, not institutional factors.” This sentence is exaggerated when we know there is a meta-analysis on the subject (Wasson et al, 2016 Association Between Learning Environment Interventions and Medical Student Well-being A Systematic Review).”

“On interventions to improve student well-being, it should be noted that most of the studies are of low quality. In addition, the effectiveness of individual interventions tends to disappear after 6 months unlike institutional interventions (Frajerman, 2020 Which interventions improve the well-being of medical students? A review of the Literature)”

We appreciate the reviewer for highlighting these studies on institutional factors associated with medical student wellbeing. We have revised this paragraph in our introduction to reflect the current state of research more accurately. Specifically, we state that institution-created wellbeing resources have been studies but that there is considerably variability in effectiveness to said programs, the studies are often of poor quality, and that the benefits may not be enduring.

Methods

3. Research Protocol

“Even if protocol was described in a previous article, authors have to put the bases in this article: the period of the study and how the questionnaire was sent to justify absence of response rate”

The manuscript has been updated to provide a brief description of the survey distribution process.

4. Individual Characteristic Clarification

“It is not clear to me whether "non male gender" represents women or women + transgender and non-binary. If it's just women, you better write it clearly.”

In our survey, we asked students how they defined their gender identity and include transgender and other as responses as we felt that this was a unique student population that was important to identify and study. In total, 9 students identified as transgender and 25 students as other. Given that these numbers were too small to include as a stand-alone category, we instead created a non-male category that included female, transgender and other. This has been more explicitly stated in the manuscript.

“The cut off for age seems strange and very high to me: why not have done 3 groups as for the course. This would be all the more interesting for multivariate analyzes”

This is an excellent point raise by the reviewer and we apologize for not providing more context. The vast majority of medical students enrolled directly out of college, which places them at 22-23 years-old when they start and 26-27 when they finish. While our original survey did look at several other age groups (<21, 22-24, 25-27, 28-31, >32), we found that there was considerable co-variability with phase in medical school in our multivariate model. Given that there is prior data on the association of phase in medical school and medical student wellbeing, we opted to eliminate the age variable as a cofounder and instead, just segregate for an older student population (>28 years-old). 

“Marital status is also strange: being in a couple is protective, but you can be in a couple without being married.”

We appreciate the reviewer addressing the point and agree. We appreciate that students can be in a relationship and not be married, which is arguably the factor that is protective for student wellbeing. In our original survey, we did not appreciate this difference and therefore, only asked our respondents about marital status and not simply relationship status. We are currently working on a project specifically looking at the impacts of medical student debt on wellbeing and have revised our survey to include questions about relationship status.

Results

5. Multivariate analysis: Medical school characteristics and wellbeing resources.

“Before performing the multivariate analyzes, it would be interesting to look at certain correlations, in particular between MEDICAL SCHOOL CHARACTERISTICS and WELLBEING RESOURCES.”

We have performed the suggested analysis and found that public universities and medium class size were negatively associated with high (>/=3) wellbeing resource availability while lower average tuition and higher faculty support were positively associated. These findings have been included in the manuscript.

6. Change in Wellbeing Scores

“The part “change in wellbeing since medical school” in figures 2 and 3 seems very questionable to me because it mixes up students of all levels. A student who feels very bad in 2nd year and only bad in 3rd year to consider that there has been an improvement, but if, on the other hand, they are all asked to recall closely to before entering medicine, there is an obvious recall bias related to their seniority. It would be more judicious to stratify by level of course or else to leave nothing.”

The data collected from the survey questions, “how do you feel your well-being has changed in the following domains since beginning medical school” is subject to recall bias. However, in this manuscript, this data is not being used as the sole measure of wellbeing. It is being used in addition to the Medical School Wellbeing Index and Leave of Absence rates, both of which are also subject to recall bias (asking those questions after coming out of a particularly challenging period of medical school, for example). The MS-WBI data and leave of absence data was not stratified based on medical school phase and we do not feel that it is necessary to do so for the change in wellbeing data. We feel that the most important finding in this collection of data is that on 3 different wellbeing metrics, wellbeing scores are better when more wellbeing resources are offered at a school.

Discussion

7. Finding specificity

“The risk factors identified are not specific to medical universities.”

We agree with the reviewer that these findings may not be specific to medical universities but do not agree with the relevance of this statement. Depression, anxiety, and suicides are high in many doctoral programs and professional schools. We are unaware of any study directly comparing rates between professional schools, but most published studies are on medical students. The findings in this manuscript that call for a variety of wellbeing resources, improving faculty supportiveness and directly listening to students’ input on wellbeing resources are meant to be broad and easy adapted by medical schools throughout the country, regardless of size or funding availability.

8. Wording

Authors wrote "especially in light of recent research which suggests that additional time on clinical rotations does not necessarily translate to a better education [51, 52]." You cant's use the word recent for studies published in 2006 and 2001

The manuscript has been updated to incorporate these changes.

Reviewer #2: 

“There is lack of Qualitative data, authors should clarify this point.”

We disagree with the statement that there is a lack of qualitative data in this manuscript. In our paper, we qualitatively analyzed free text analysis from >1,000 survey respondents. We inductively coded the free text and categorized the codes by domain and themes (Figure 3). This data is helpful for understanding a students’ perspective on usefulness of wellbeing resources and suggest that students want to see more resources for mental health services, specifically have been access and time available to utilize these services. This data is also useful in helping guide institution on which wellbeing resource are most efficacious. For instance, if an institution was determining if it would be better to put resources into providing a stipend for educational resources or better access and availability of fitness programs for students, more students thought fitness resources would be more beneficial to them. We hope that we have clarified this point for the reviewer.

---

## [Editor Report · Decision Letter 1]

21 Dec 2021

PONE-D-20-40527R1

Medical Schools Contribute to Student Distress The Impact of Faculty Support and Wellbeing Resource Availability

PLOS ONE

Dear Dr. Langness,

Thank you for submitting your manuscript to PLOS ONE. After careful consideration, we feel that it has merit but does not fully meet PLOS ONE’s publication criteria as it currently stands. Therefore, we invite you to submit a revised version of the manuscript that addresses the points raised during the review process.

The Academic Editor for your manuscript and the Deputy Editor-in-Chief are satisfied with the revisions to your manuscript made in response to the reviewers' comments. However, following editorial discussion, we consider that additional revisions are required to meet PLOS ONE's 4th publication criterion, requiring that conclusions are presented in an appropriate fashion and are supported by the data.

As noted, the study described in the manuscript is observational. As such it is not appropriate to make statements that imply causation; an interventional study would be required to make conclusions of this nature. A number of revisions are provided below to address these concerns. Could you please update the manuscript accordingly?

We look forward to receiving your revised manuscript.

Kind regards,

Kamran Sattar

Academic Editor

PLOS ONE

with

George Vousden

Deputy Editor-in-Chief

PLOS ONE

Journal Requirements:

Additional Editor Comments (if provided):

Revisions requested:

Title: Suggest to change “Medical Schools Contribute to Student Distress: The Impact of Faculty Support and Wellbeing Resource Availability” to “Risk factors associated with student distress in medical schools: Associations with faculty support and wellbeing resource availability”Line 50: Suggest to change to “Schools can help mitigate medical student distress by improving faculty support and offering more and varied wellbeing resources” to “Improving faculty support and offering more and varied wellbeing resources may help to mitigate medical student distress.”Line 109: Suggest to change “…determine the impact of institutional investment in wellbeing resources on medical student wellbeing…” to “…determine whether there are associations between institutional investment in wellbeing resources and medical student wellbeing…”Line 345-359: The first paragraph of the discussion needs to be rephrased to avoid using language implying causation, e.g. “we found that medical schools both contribute to and help mitigate against severe distress”.Line 362: Suggest to change “In our study, having more wellbeing resources at one’s school was protective against severe distress. Interestingly, the availability of multiple wellbeing resources mitigated severe distress regardless of whether students utilized the resource.” to “In our study, having more wellbeing resources at one’s school was associated with less severe distress. Interestingly, the availability of multiple wellbeing resources moderated the relationship with severe distress regardless of whether students utilized the resource.”Line 477-483: Revise conclusion to avoid language implying causation, e.g. “medical schools themselves play a role in medical student wellbeing”
---

## [Author Response · Author response to Decision Letter 1]

25 Feb 2022

1. Title: Suggest to change “Medical Schools Contribute to Student Distress: The Impact of Faculty Support and Wellbeing Resource Availability” to “Risk factors associated with student distress in medical schools: Associations with faculty support and wellbeing resource availability” 

- Edited as suggested.

2. Line 50: Suggest to change to “Schools can help mitigate medical student distress by improving faculty support and offering more and varied wellbeing resources” to “Improving faculty support and offering more and varied wellbeing resources may help to mitigate medical student distress.” 

- Edited as suggested.

3. Line 109: Suggest to change “…determine the impact of institutional investment in wellbeing resources on medical student wellbeing…” to “…determine whether there are associations between institutional investment in wellbeing resources and medical student wellbeing…” 

- Edited as suggested.

4. Line 345-359: The first paragraph of the discussion needs to be rephrased to avoid using language implying causation, e.g. “we found that medical schools both contribute to and help mitigate against severe distress”. 

- Added “may” to avoid implying causation.

5. Line 362: Suggest to change “In our study, having more wellbeing resources at one’s school was protective against severe distress. Interestingly, the availability of multiple wellbeing resources mitigated severe distress regardless of whether students utilized the resource.” to “In our study, having more wellbeing resources at one’s school was associated with less severe distress. Interestingly, the availability of multiple wellbeing resources moderated the relationship with severe distress regardless of whether students utilized the resource.” 

- Edited as suggested.

6. Line 477-483: Revise conclusion to avoid language implying causation, e.g. “medical schools themselves play a role in medical student wellbeing” 

- Added “may” to avoid implying causation.

---

## [Editor Report · Decision Letter 2]

10 Mar 2022

Risk Factors Associated with Student Distress in Medical Schools: Associations with Faculty Support and Availability of Wellbeing Resources

PONE-D-20-40527R2

Dear Dr. Langness,

We’re pleased to inform you that your manuscript has been judged scientifically suitable for publication and will be formally accepted for publication once it meets all outstanding technical requirements.

Kind regards,

Kamran Sattar

Academic Editor

PLOS ONE
---

## [Editor Report · Acceptance letter]

31 Mar 2022

PONE-D-20-40527R2 

Risk Factors Associated with Student Distress in Medical School: Associations with Faculty Support and Availability of Wellbeing Resources 

Dear Dr. Langness:

I'm pleased to inform you that your manuscript has been deemed suitable for publication in PLOS ONE. Congratulations! Your manuscript is now with our production department. 

Kind regards, 

on behalf of

Dr. Kamran Sattar 

Academic Editor

PLOS ONE